# Metabolomic Analysis of Key Metabolites and Their Pathways Revealed the Response of Alfalfa (*Medicago sativa* L.) Root Exudates to *rac*-GR24 under Drought Stress

**DOI:** 10.3390/plants12051163

**Published:** 2023-03-03

**Authors:** Yuwei Yang, Mingzhou Gu, Junfeng Lu, Xin’e Li, Dalin Liu, Lin Wang

**Affiliations:** 1College of Animal Science and Technology, Yangzhou University, Yangzhou 225009, China; 2College of Horticulture and Plant Protection, Yangzhou University, Yangzhou 225009, China; 3Institutes of Agricultural Science and Technology Development, Yangzhou University, Yangzhou 225009, China; 4Joint International Research Laboratory of Agriculture and Agri-Product Safety, The Ministry of Education of China, Yangzhou University, Yangzhou 225009, China

**Keywords:** alfalfa, *rac*-GR24, metabolomic pathway, root exudate

## Abstract

The *rac*-GR24, an artificial analog of strigolactone, is known for its roles in inhibiting branches, and previous studies have reported that it has a certain mechanism to relieve abiotic stress, but the underlying metabolic mechanisms of mitigation for drought-induced remain unclear. Therefore, the objectives of this study were to identify associated metabolic pathways that are regulated by *rac*-GR24 in alfalfa (*Medicago sativa* L.) and to determine the metabolic mechanisms of *rac*-GR24 that are involved in drought-induced root exudate. The alfalfa seedling WL-712 was treated with 5% PEG to simulate drought, and *rac*-GR24 at a concentration of 0.1 µM was sprayed. After three days of treatment, root secretions within 24 h were collected. Osmotic adjustment substances and antioxidant enzyme activities were measured as physiological indicators, while LS/MS was performed to identify metabolites regulated by *rac*-GR24 of root exudate under drought. The results demonstrated that *rac*-GR24 treatment could alleviate the negative effects from drought-induced on alfalfa root, as manifested by increased osmotic adjustment substance content, cell membrane stability, and antioxidant enzyme activities. Among the 14 differential metabolites, five metabolites were uniquely downregulated in plants in *rac*-GR24 treatment. In addition, *rac*-GR24 could relieve drought-induced adverse effects on alfalfa through metabolic reprogramming in the pathways of the TCA cycle, pentose phosphate, tyrosine metabolism, and the purine pathway. This study indicated that *rac*-GR24 could improve the drought resistance of alfalfa by influencing the components of root exudates.

## 1. Introduction

Drought is a global issue, with frequent occurrence of extreme weather around the world [1]. Physiological mechanisms of plant responses to drought include effective biomass allocation changes [2], strong osmotic regulation capacity [3], low membrane lipid peroxidation, and low ROS accumulation levels [4]. In fact, the response of plants to drought stress is a complex biological process involving many metabolic changes [5,6]. It is estimated that there are about 200,000 to 1 million metabolites in all plants [7], including primary metabolites, such as carbohydrates, lipids, amino acids, nucleic acids, and organic acids, etc. The metabolites in plants also include secondary metabolites closely related to plant stress defense, such as alkaloids, phenols, quinones, flavonoids, and terpenes [8]. Previous research on the effects of drought stress on plant metabolomics has revealed that the general metabolic changes in plants in response to drought are significantly increases in amino acids, organic acids, sugars, and polyols [9,10]. These compounds can be regulated osmotically through scavenging ROS, protecting cellular components, or ensuring membrane lipid stability [11].

The root system has abilities to improve the ecological environment and maintain water in soil [12]. Newly created lateral roots can efficiently transport water to the main root in the drought environment [13]. Compared with other crops, *Medicago sativa* is a kind of best pasture for feed that is among the most widely cultivated forages in the world [14]. Alfalfa has a strong deep root system that can acquire and absorb deep soil moisture, which is widely considered to be one of the main reasons for its high drought resistance. Nonetheless, drought is a major environmental factor which can cause damage to the growth and yield of alfalfa [15]. The root cells will lose water and shrink when the water absorbed by the roots of alfalfa is limited.

The existing measures to improve plant stress resistance include breeding, adding exogenous plant hormones, or genetic engineering. However, adding exogenous hormones is one of the most widely used methods. Among plant hormones, strigolactone (SL) has been identified as a type of plant hormone that can certainly improve stress resistance of plants. Previous studies have also reported that one of the artificial analogs of SL, GR24, can improve the heat tolerance of lupine [16] and the NaCl tolerance of maize [17]. However, to our knowledge, understanding of the metabolite changes associated with the mitigation effect of GR24 on alfalfa grown under drought stress remains fragmentary. In particular, the effect of GR24 on the roots and exudates of alfalfa under drought stress is still unclear.

Root exudates from plants undergoing abiotic stress defense contain compounds (amino acids, carbohydrates, peptides, and phenolic compounds) with osmotic regulation and antioxidant capacity. More and more studies have found that root exudates are important signal substances, among which ethylene, salicylic acid, and jasmonic acid are effective signal substances in the rhizosphere, which can transmit information in plant–plant interactions, stimulating intraspecific and interspecific subsurface responses [18]. In our previous study, transcriptomics [19] and chlorophyll a (Chl a) OJIP fluorescence [20] were used to analyze the mechanism of GR24 in alleviating the growth of alfalfa under drought stress, but the study materials were leaves. In order to study the alleviating effect of *rac*-GR24 on alfalfa growth under drought stress, we investigated the mechanism of *rac*-GR24 alleviating alfalfa growth under drought stress in this study by analyzing the effect of *rac*-GR24 on alfalfa root exudates.

## 2. Methods and Materials

### 2.1. Material Cultivation

At Yangzhou University, this experiment was carried out by utilizing WL-712 (*Medicago sativa* L.) as the plant material. The Zheng Dao Company in Beijing donated the WL-712 seeds, which were chosen for their consistent size. WL-712 alfalfa seeds were grown in germination boxes for 10 days and then transferred to plastic containers. All 16 plastic containers (475 mm × 375 mm × 165 mm) were filled with 28 L full Hoagland solution, and 20 selected seedlings were sown in each container. The Hoagland solution was changed every 10 days. All of the containers were kept in a greenhouse at a temperature of 25/18 °C (day/night) with a 16-h photoperiod.

### 2.2. Treatments

Seedlings were given 4 different treatments for 3 days after 30 days of cultivation, according to the following: (1) The untreated samples were Control; (2) SL was sprayed with 0.1 μM *rac*-GR24; (3) D treatment was planted with 5% PEG; (4) DSL was based on D treatment and sprayed with 0.1 μM *rac*-GR24. A 10 μM measure of stock solution was used to prepare *rac*-GR24 solution, and 3 mg *rac*-GR24 (Chiralix, Nijmegen, The Netherlands) was dissolved in 500 μL of acetone and diluted in 1 L of distilled water. At each time, 50 mL of solution was used for foliar application of each plant. After 3 days of treatment, the roots were washed with sterile water and put into a sterile centrifuge tube filled with 30 mL of sterile water as root exudates for 24 h later. Therefore, root samples were collected from plants three days after osmotic stress exposure for further physiological analysis.

### 2.3. Determination of Triphenyltetrazolium Chloride (TTC), Soluble Sugar (SS), Soluble Protein (SP), and Malondialdehyde (MDA)

According to Clemensson-Lindell [21], the TTC reduction strength was measured by the TTF reaction method. The content of soluble sugar was determined according to Buysse and Merckx [22]. The content of soluble protein was determined by the method of Bradford [23]. According to Liu [24], the MDA content was determined using the TBA reaction.

### 2.4. Determination of Strigolactone, O_2_^−^, and Enzyme Activity

The content of strigolactone in roots was determined using a plant ELISA kit (Jiangsu Meimian Industrial Co., Ltd., Yancheng, China) and the content of O_2_^−^ was measured by a superoxide anion content detection kit (Beijing Solarbio Science & Technology Co., Ltd., Beijing, China, BC1290). The activity of catalase (CAT) was measured following the method described by Blume and McClure [25], and the peroxidase (POD) activity was calculated by the method of Ranieri [26].

### 2.5. Untargeted Metabolomics Analysis

The metabolites present in the samples of D and DSL treatment were determined using LC–MS at BioNovoGene Co., Ltd. (Suzhou, China). The samples of root exudate solution were thawed at 4 °C and mixed for 1 min after thawing. An appropriate amount of sample was transferred accurately into a 2 mL centrifuge tube, 500 µL methanol (stored at −20 °C) was added into the sample tube, and the contents were dried and vortexed for 1 min. A 150 µL measure of 2-Amino-3-(2-chloro-phenyl)-propionic acid (4 ppm) solution was added to prepare with 80% methanol water (stored at −20 °C) to redissolve the sample. A 0.22 μm membrane was used to filter the supernatant and LC–MS detection was transferred into the detection bottle.

For LC analysis, the Vanquish UHPLC system (Thermo Fisher Scientific, Waltham, MA, USA) was used. ACQUITY UPLC ^®^ HSS T3 (150 × 2.1 mm, 1.8 µm) (Waters, Milford, MA, USA) was used to carry out chromatography. The flow rate was 0.25 mL/min and 2 μL was set as the injection volume. Mass spectrometric detection of metabolites was performed by using Q Exactive Focus (Thermo Fisher Scientific, Waltham, MA, USA) with an ESI ion source. Robust LOESS signal correction (QC-RLSC) was applied for data normalization to correct for any systematic bias. Normalization was based on the proportion conversion of quantitative values of a single metabolite and the sum of quantitative values of all metabolites in the sample. After normalization, only ion peaks with relative standard deviations less than 30% in QC were kept, to ensure proper metabolite identification.

### 2.6. RNA Extraction and Gene Expression Analysis

After root exudates were collected, 0.05 g of D and DSL alfalfa roots were taken for RNA extraction. Total RNA was extracted by FastPure Universal Plant Total RNA Isolation Kit (catalogue No. RC411, Vazyme, Nanjing, China). In the following procedure, the relative expression levels of differentially expressed genes (DEGs) were measured by qRT-PCR. To summarize, cDNA was synthesized by a kit from Vazyme in Nanjing, China, called HiScript III RT SuperMix. The QuantStudio3 Real-Time PCR system and 20 μL volumes with SYBR as a quantitative dye (Vazyme, Nanjing, China) were used to perform qRT-PCR. The β-actin gene was chosen as a reference gene. Twenty-six selected genes with information are listed in Appendix A
Table A1. Four biological replicates and three technical replicates were used independently for the qRT-PCR analysis.

### 2.7. Statistical Analysis

The R package ropls was used to analyze processed data after being normalized to the total peak intensity. After scaling data, models were built on principal component analysis (PCA). The metabolic profiles could be visualized as a score plot, where each point represents a sample. To identify the contributable variable for classification, the *p*-value, variable importance projection (VIP), and fold change (FC) were used. Metabolites with *p* < 0.05 and VIP values > 1 were considered statistically significant. The identified metabolites in metabolomics were then mapped to the KEGG pathway for biological interpretation of higher-level systemic functions. The metabolites and corresponding pathways were visualized using the KEGG Mapper tool. In the growth and physiological analysis of alfalfa seedlings under drought stress, each test was performed in triplicate, and data were expressed as the mean (with SD) of four independent replicates. In SPSS 23.0 software, one-way ANOVA (*p* < 0.05) was used to perform statistical analyses.

## 3. Results

### 3.1. Morphological Response to Drought Stress in Alfalfa Root

The length of alfalfa roots was significantly increased by 54.9% in DSL treatment, comparing with D, while the weight of roots was decreased by 41.7%. In addition, the TTC reaction strength was dramatically increased by 61.2% in DSL treatment, compared to D (Figure 1C).

### 3.2. SL Response to Drought Stress in Alfalfa Root

The content of SL in alfalfa roots was sharply increased by 63.8% in SL treatment compared to the control group, whereas the SL content was increased by 52.1% in DSL treatment (Figure 2).

### 3.3. Osmotic Substance Response to Drought Stress in Alfalfa Root

Compared to D, the content of soluble sugar in DSL treatment alfalfa roots was significantly increased by 28.0%, while the content of soluble protein was increased by 7.6%, not significantly (Figure 3).

### 3.4. Membrane Lipid Peroxidation Response to Drought Stress in Alfalfa Root

The content of MDA in SL treatment was decreased compared to Control, while that in DSL treatment was significantly decreased by 36.1% (D vs. DSL) (Figure 4).

### 3.5. Antioxidant Response to Drought Stress in Alfalfa Root

The content of O_2_^−^ was significantly decreased under the DSL treatment contrasted to the D treatment. The content of O_2_^−^ was significantly decreased by 48.0%, whereas the activity of POD was increased by 7.1% (D vs. DSL). The main antioxidants regulating ROS to reduce cell damage were CAT and POD activities. Compared to the control group, all *rac*-GR24 treatments showed an increase in antioxidant enzyme activities in both SL and DSL treatment. The activities of POD and CAT were increased by 19.3% and 37.2%, respectively, comparing SL and Control, and by 7.1% and 10.2%, respectively, comparing D and DSL (Figure 5).

### 3.6. Metabolic Profiling

The samples from different treatments showed distinct separation, suggesting significant changes in metabolites in those samples. In total, 261 metabolites were detected from the six alfalfa root exudate samples (D vs. DSL) by the LS/MS system. The detected metabolite content data were normalized. The metabolite accumulation pathways of different samples were clustered. Among all metabolites identified, 126 metabolites were upregulated and 131 metabolites were downregulated (D vs. DSL). The accumulated metabolites were considered differentially accumulated in a statistically significant way by a VIP (variable importance on projection) value > 1 and *p* < 0.05. Among 261 metabolites, 14 differentially accumulated metabolites were identified (Figure 6A). The contents of ethyl benzoate, phenylacetaldehyde, 4-Hydroxybenzoate, p-Octopamine, and L-Malic acid were decreased in DSL treatment, but the contents of paclobutrazol, uric acid, 4-Hydroxy-2-quinolone, and 3-Methoxytyramine were significantly increased (Figure 6B).

### 3.7. KEGG Enrichment Analysis

To further analyze the differentially accumulated metabolites which were related to improving alfalfa drought tolerance, KEGG pathway analysis was carried out for all differentially accumulated metabolites in alfalfa. KEGG analysis assigned 14 differentially regulated metabolites to 12 metabolic pathways (Figure 7). L-Malic acid, 3-Methoxytyramine, and 4-Hydroxybenzoate were involved in the most KEGG pathways (four and two pathways, respectively). In particular, KEGG analysis indicated that two differentially accumulated metabolites (rosmarinic acid and 3-Methosytyramine) were involved in the tyrosine metabolism pathway. In addition, DEMs mainly enriched in the citrate cycle and carbon fixation in photosynthetic organism pathways. Generally, these main enriched metabolic pathways can be divided into six classes: amino acid metabolism, biosynthesis of other secondary metabolites, carbohydrate metabolism, energy metabolism, metabolism of cofactors, and vitamins and nucleotide metabolism.

### 3.8. Genes Related to rac-GR24-Induced in Alfalfa Roots under Drought Stress

Based on KEGG analysis results, the relative expressions of 26 genes were selected to measure by qRT-PCR (Figure 8A). Figure 8B shows metabolic pathways for the differential metabolites in DSL treatment in comparison to those in D treatment of alfalfa root exudates. Twelve associated genes, including *Glucose-6-phosphate 1-dehydrogenase 2* (*G6PD 2*), *Probable 6-phosphogluconolactonase 1* (*6PGL 1*), *Ribulose-phosphate 3-epimerase* (*RP3E*), *Fructose-bisphosphate aldolase 1* (*Fba 1*), and *Isocitrate dehydrogenase* (*IDH*), were upregulated in the TCA cycle and pentose phosphate pathway in DSL treatment. Only the relative expression of *Glucose-6-phosphate isomerase 1* (*G6PI 1*) was downregulated. The relative expressions of *Primary amine oxidase* (*PAO*) and *DHBP synthase* were elevated, whereas *GMP synthase* was downregulated in the tyrosine metabolism pathway. Additionally, seven genes were upregulated and three genes were downregulated among the 10 selected genes that are parts of the purine metabolism and folate production pathway. Since most of the structural genes involved in the pathways were dramatically downregulated while *G6PI 1* and *GMP synthase* were significantly downregulated in DSL treatment, the TCA cycle, pentose phosphate metabolism, and tyrosine metabolism in alfalfa roots were all activated. Interestingly, GDP synthesis is activated in the purine metabolic pathway, while genes involved in urate synthesis at the end of pathway are inhibited. These results may suggest that pentose phosphate and tyrosine metabolism in alfalfa under drought stress is a *rac*-GR24-induced mitigation mechanism.

## 4. Discussion

Serious browning occurred in the underground part of alfalfa exposed to drought stress, and results revealed the biochemical changes (mainly including antioxidant enzyme activity and osmotic adjustment substance) within WL-712 alfalfa roots in response to drought stress in DSL treatment (Figure 1). The roots showed relatively significant changes in the membrane, antioxidants, and osmolytes under drought stress. The TTC reaction strength of alfalfa roots was significantly decreased under drought stress and increased in DSL treatment, which indicated that *rac*-GR24 improved antioxidant enzyme activities of alfalfa root (Figure 1C). Stress-treated rice and cucumber have shown a boost in the content of MDA and O_2_^−^ [27,28]. In this study, the decreasing of MDA and O_2_^−^ contents were also observed in comparison with D and DSL, which may suggest that osmolytes could protect the membranes of plant cells, resulting in increased membrane stability and balanced osmotic pressure (Figure 4 and Figure 5). The contents of SP and SS were increased in a boost, which indicated that *rac*-GR24 at a concentration of 0.1 µM could modulate the response to drought stress by inducing the accumulation of osmotic substances to reduce water potential in order to maintain the water absorption capacity of alfalfa (Figure 3). As a defense strategy, the enzymatic antioxidant system was used to clear ROS and increase the plant’s resilience to stress [29]. POD and CAT (Figure 5) may therefore be key players in the detoxification of ROS produced as a result of drought stress in alfalfa roots. Additionally, by oxidizing amino acids and proteins, ROS can harm cells and the photosynthetic apparatus [30].

In this study, untargeted metabolomics studies showed that 12 key metabolic pathways involving 14 metabolites were the metabolic response mechanism of drought-stress resistance of alfalfa in DSL treatment. These metabolites are involved in the TCA cycle, pentose phosphate, tyrosine metabolism, betalain biosynthesis, purine metabolism, and folate biosynthesis. The pentose phosphate pathway is a direct oxidative decomposition pathway of glucose, which is closely related to aerobic respiration of cells [31]. As one of the major upregulated differential metabolites (D vs. DSL), 6-phosphogluconic acid is significantly enriched in pentose phosphate metabolism. According to the results of qRT-PCR analysis, the conversion pathway of β-D-Glucose-6P to α-D-Glucose-6P was inhibited by *rac*-GR24, but DSL treatment catalyzed the conversion of β-D-Glucose-6P to D-Glucono-1,5-lactone-6P, and finally enhanced the pentose phosphate pathway, which enabled the plants to carry out normal aerobic respiration under drought stress, and the TTC activities in roots indeed significantly increased (D vs. DSL). The TCA cycle is a series of catabolic reactions that occurs in the mitochondria. It combines the carbon dioxide produced by the oxidation of pyruvate and malate with the production of nicotinamide adenine dinucleotide (NADH) to create energy for the respiratory chain to oxidize [32]. Differential metabolism analysis showed that the direct regulation of the TCA cycle in plants was provided by L-malic acid, which might be due to DSL treatment can enhance plant respiration and make plants obtain more energy, indicating that the TCA cycle rate in the roots of alfalfa in DSL treatment was increased. In addition, carbohydrates are the main energy storage substances and can be regulated reacting to drought stress, as carbohydrate content has been regarded as an indicator of plant physiological state [33]. In this study, soluble sugar, as carbohydrate, was involved in the TCA cycle and significantly increased in roots under drought stress and *rac*-GR24 treatment, implying that, by increasing carbohydrate content, roots could accelerate energy support and produce more drought-stress-related proteins, and thus alleviate drought stress. At the same time, the expressions of genes regulating carbohydrates were significantly upregulated, which further supports this result.

Nucleotide biosynthesis and degradation are two of the metabolic processes in responding to drought of Arabidopsis [34] and rice [35]. Recent studies have indicated that the purine metabolism pathway is a crucial way for *Dendrobium sinense* [36] and *Dendrobium wangliangii* [37] to adapt to drought stress. Previous studies showed that guanine deficiency causes xanthine and hypoxanthine to replace guanine in RNA and DNA synthesis, leading to metabolic system disorders [38]. In this study, DSL treatment promoted the conversion of GDP to GTP, which could speed up the energy conversion process within cells. Consistent results at the gene level show that *Pyruvate kinase* (*PK*) and *Ribonucleoside-diphosphate reductase small chain A* (*RNR2*) that control DNA replication were upregulated in DSL treatment (Figure 8A). This indicates that *rac*-GR24 could enable plants to resist the system disorder caused by drought by enhancing the DNA replication process under stress. In addition, the way from GDP to XMP was inhibited, while the branch of IDP was promoted, which caused that inosine was produced. The urate in root exudate was significantly increased in DSL treatment. According to qRT-PCR result, the expression of *Hypoxanthine-guanine phosphoribosyltransferase* (*HGPRT*) related to xanthine metabolite was upregulated, while the expression of *Hydroxyisourate hydrolase* (*HIU hydrolase*) related to urate was downregulated (Figure 8A). These results suggest that *rac*-GR24 can improve the drought tolerance of alfalfa by promoting the synthesis of xanthine, inosine, and uric acid, reducing allantoin and enhancing purine metabolism, while restoring the normal metabolic system.

As upregulated metabolites, rosmarinate and 3-Methoxytyramine were mainly enriched in tyrosine metabolism and betaine metabolism pathways. Tyrosine is a biosynthetic precursor of tocopherols, plastoquinone, and ubiquinone, all of which are required by plants [39]. Together with tocotrienols, tocopherols make up the class of lipid-soluble antioxidants known as tocochromanols, which have physiological functions for plants other than antioxidation [40]. Under drought stress, the tyrosine metabolic pathway is activated, and the production of rosmarinate effectively improves the antioxidant activity of plants. In terms of physiological indicators, the activities of POD and CAT were enhanced, and the content of O_2_^−^ was also significantly decreased. A significant increase in the content of rosmarinate in alfalfa root exudate in DSL treatment indicates that the plant’s antioxidant activity remains high, which indicates that *rac*-GR24 could effectively improve the activities of antioxidant enzymes in plant cells to eliminate ROS.

## 5. Conclusions

In conclusion, *rac*-GR24 could alleviate the damage caused by drought stress on alfalfa roots by regulating a series of metabolic pathways (Figure 9). The mechanism of *rac*-GR24 alleviating drought damage to alfalfa is related to respiration energy, antioxidant substances and osmoregulatory substances in the TCA cycle, tyrosine metabolism, and purine metabolism. *rac*-GR24 could improve plant respiration through the pentose phosphate pathway and TCA cycle, promote purine metabolism, increase urate content, and strengthen the DNA replication process to resist DNA damage caused by drought. In addition, *rac*-GR24 could improve the content of rosmarinic acid to increase the activities of antioxidant enzymes and remove ROS. This study indicated that *rac*-GR24 improves the drought resistance of alfalfa by affecting the components of root exudates. The relationship between root exudates and soil microorganisms will be investigated in a further study.

## Figures and Tables

**Figure 1 plants-12-01163-f001:**
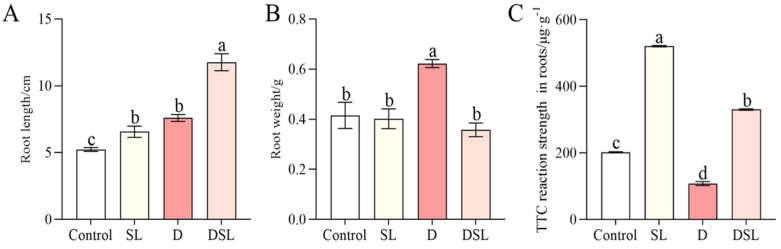
Morphological changes in alfalfa root in response to different treatments include root length (**A**), root weight (**B**), and TTC (**C**). Results are presented as the mean of four independent experiments ± standard error. ^a–d^ Different superscripts mark significant differences between the treatments (*p* < 0.05).

**Figure 2 plants-12-01163-f002:**
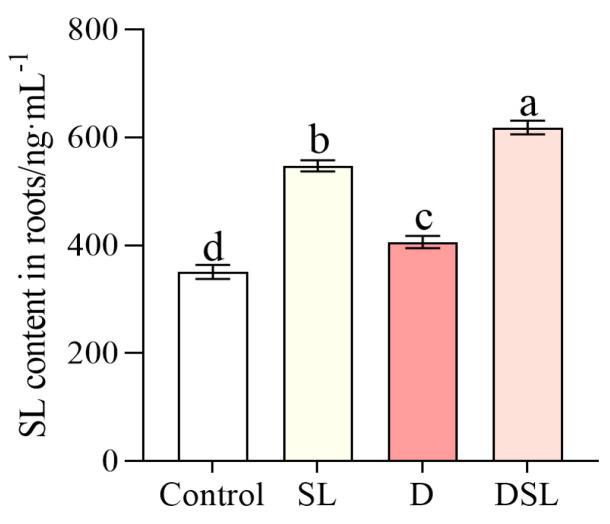
SL content of alfalfa root in response to different treatments. Results are presented as the mean of four independent experiments ± standard error. ^a–d^ Different superscripts mark significant differences between the treatments (*p* < 0.05).

**Figure 3 plants-12-01163-f003:**
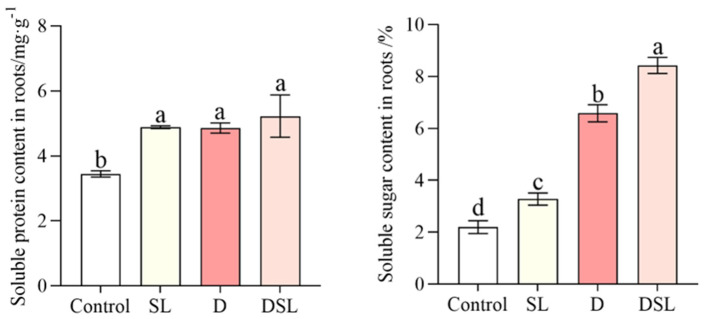
Soluble sugar and soluble protein contents of alfalfa root in response to different treatments. Results are presented as the mean of four independent experiments ± standard error. ^a–d^ Different superscripts mark significant differences between the treatments (*p* < 0.05).

**Figure 4 plants-12-01163-f004:**
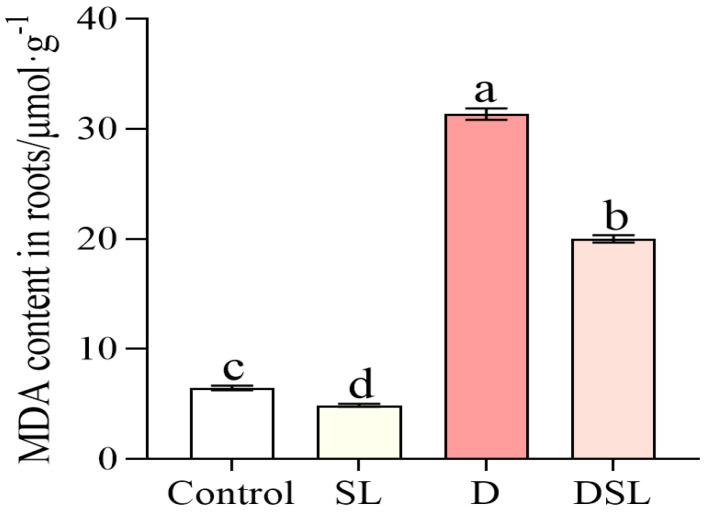
MDA contents of alfalfa root in response to different treatments. Results are presented as the mean of four independent experiments ± standard error. ^a–d^ Different superscripts mark significant differences between the treatments (*p* < 0.05).

**Figure 5 plants-12-01163-f005:**
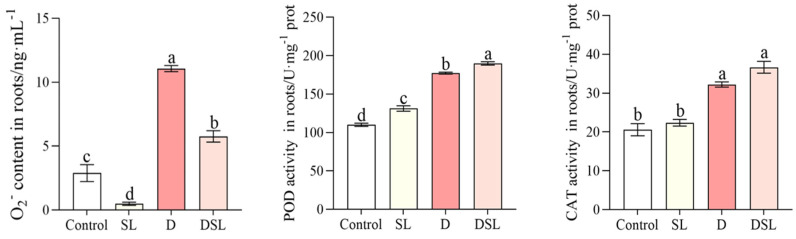
The content of O_2_^−^, POD, and CAT activities of alfalfa root in response to different treatments. Results are presented as the mean of four independent experiments ± standard error. ^a–d^ Different superscripts mark significant differences between the treatments (*p* < 0.05).

**Figure 6 plants-12-01163-f006:**
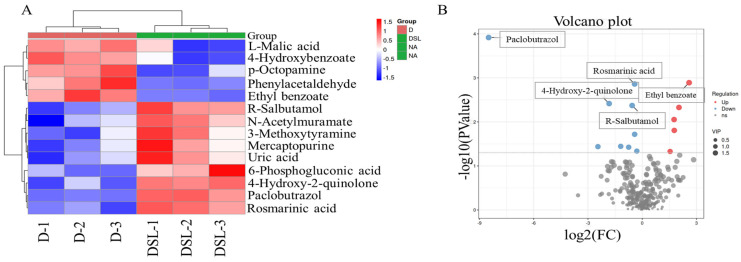
Heat map showing fold changes for metabolites responsive to *rac*-GR24 treatment, with increases (upregulation showing in red) or decreases (downregulation showing in blue) in the root exudates in drought stress compared to the *rac*-GR24 treatment (**A**). The volcano maps show significantly upregulated or downregulated metabolites (**B**).

**Figure 7 plants-12-01163-f007:**
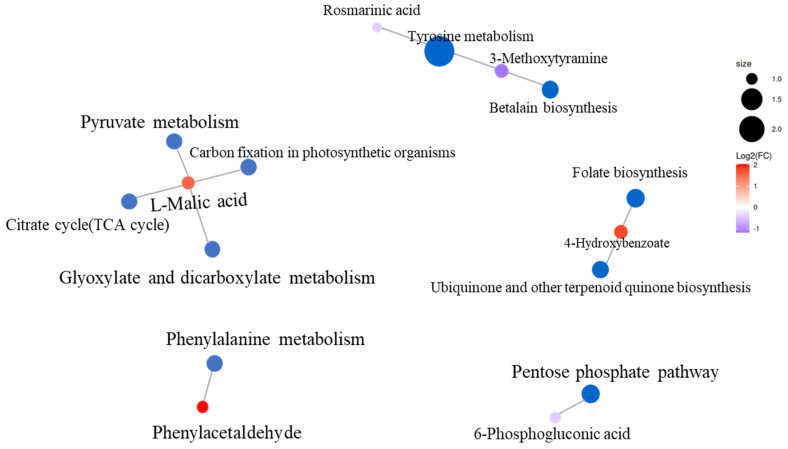
The network diagram shows the KEGG pathway where differential metabolites are mainly enriched.

**Figure 8 plants-12-01163-f008:**
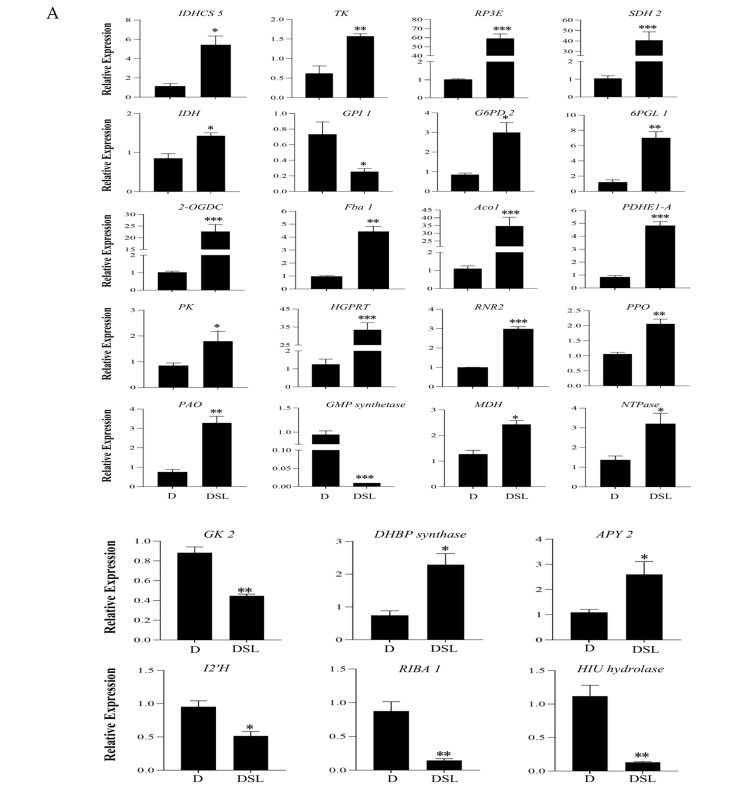
Relative expressions of 26 selected genes (**A**) and metabolic pathways for the differential metabolites in DSL treatment in comparison to those in D treatment of alfalfa root exudates (**B**). The red boxes represent those upregulated and the green boxes represent those downregulated in DSL compared to those in D. Results are presented as the mean of four independent experiments ± standard error. * *p* < 0.05 relative to drought treatment, ** *p* < 0.01 relative to drought treatment, *** *p* < 0.001 relative to drought treatment.

**Figure 9 plants-12-01163-f009:**
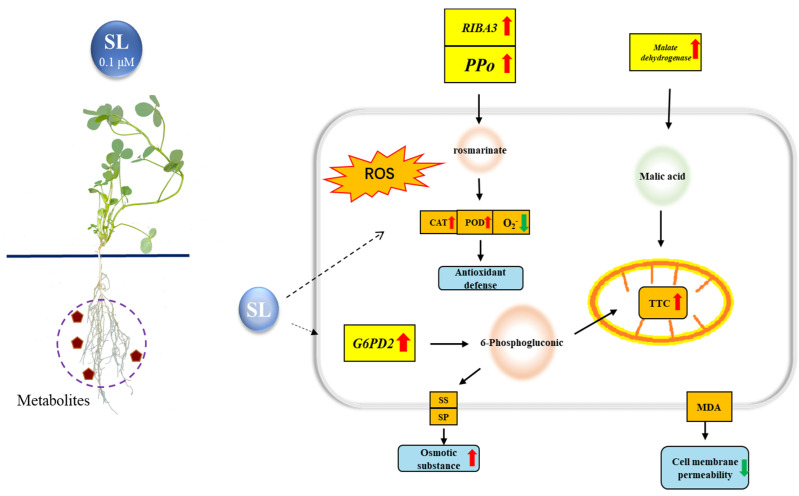
Summary of the changes in physiological characteristics, DEMs, and related genes of alfalfa roots under drought stress and with the addition of *rac*-GR24 for drought tolerance. Orange circles represent rosmarinate and 6-Phosphogluconic, which were upregulated, while the green circle represents malic acid, which was downregulated.

## Data Availability

Data are contained within the article.

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
