# Peer review of "Metabolomic Analysis of Key Metabolites and Their Pathways Revealed the Response of Alfalfa (Medicago sativa L.) Root Exudates to rac-GR24 under Drought Stress"

_plants, 2023, doi:10.3390/plants12051163_

Round 1

Reviewer 1 Report

The article is highly topical as it elaborates mechanism determining drougth stress and can support developments to increase it. The authors evidently are experts in the field and the study is based on advanced experimental study. The discussion is based on experimental results and up-to data analysis is provided. Some minor comments:

It would be good to provide formulation of aim of the study.

More details about LC/MS analysis and especially data treatment and analysis

Letters in figures are too small in comparison with the text – when possible, I suggest to increase it

Author Response

Dear Sir/Madam,

We specifically want to thank the editor and the reviewers for the comments. We appreciate the valuable suggestions provided by the editor and the reviewers. Enclosed please find a revised version of the manuscript. We have incorporated the suggestions made by the reviewers. The revised parts of the manuscript are highlighted in yellow color. Here are our detailed point-by-point responses to the comments of reviewers. We hope the revised manuscript meet the requirements for publication in Plants. We look forward to hearing from you soon.

1) It would be good to provide formulation of aim of the study.

Thank you very much for reviewing our manuscript. We added the objective of this study in Abstract section (Page 1, line 14-16) and Introduction section (Page 2, line 70-73).

2) More details about LC/MS analysis and especially data treatment and analysis

According to your valuable suggestion, we have re-written 2.5. Untargeted Metabolomics Analysis section and 2.7. Statistical analysis section, which were highlighted in yellow color (Page 3-4).

3) Letters in figures are too small in comparison with the text – when possible, I suggest to increase it

Thank you very much for your comment. We have increased the sizes of letters in figures.

Reviewer 2 Report

I had a great opportunity to review manuscript entitled: 'Metabolomics analysis of key metabolites and their pathways revealed the response of alfalfa (Medicago Sativa L.) root exudates to rac-GR24 under drought stress' which is considered for publication in Plants. The manuscript is elaborated on an interesting topic and clearly summarize new data valuable for the research community. The author has done a good job at describing the problem, the methods and the results.

GENERAL COMMENTS:
TITLE
The paper title is well stated, it is informative and concise.

ABSTRACT, INTRODUCTION
Abstract is well written with the key findings of the study. Introduction is concise, focused and informative.

MATERIAL AND METHODS
Material and research methods are presented appropriately and clearly. Experimental setup and the description in the methods section are well structured, and the statistical analysis is done alright.

RESULTS
The results obtained in this study are interesting. Results presented correctly.

DISCUSSION
In general, the discussion of results is correct. Please focus on the findings from the results in this research and finally answer the scientific problem.

CONCLUSIONS
This part needs to be improved. In the Conclusion I suggest to writing two other words on the aspect concerning the aspects where the future studies must be oriented.

LITERATURE
The items of literature included in the paper are rather sufficient and adequate to the subject of the paper.

The text of the manusctipt is not formatted correctly yet. The manuscript has many punctuation and syntax errors that must be corrected prior to publication- please check the whole manuscript. Most often they are related to species nomenclature, species name should be in italics. Please check the whole manuscript and correct GR24 to rac-GR24. I also recommend revision of the English language by a native speaker or a commercial entity to remove minor typos in the text.

Author Response

Dear Sir/Madam,

We specifically want to thank the editor and the reviewers for the comments. We appreciate the valuable suggestions provided by the editor and the reviewers. Enclosed please find a revised version of the manuscript. We have incorporated the suggestions made by the reviewers. The revised parts of the manuscript are highlighted in yellow color. Here are our detailed point-by-point responses to the comments of reviewers. We hope the revised manuscript meet the requirements for publication in Plants. We look forward to hearing from you soon.

Q1: This part needs to be improved. In the Conclusion I suggest to writing two other words on the aspect concerning the aspects where the future studies must be oriented.

Thank you very much for your comment. We have added the content in Conclusion section which were highlighted in yellow color (Page 11, line 350-352 ).